# The Significance of Digital Citizenship and Gender in the Relationship between Social Media Usage Time and Self-Esteem among Adolescents: A Secondary Analysis

**DOI:** 10.3390/children10091561

**Published:** 2023-09-15

**Authors:** Euna Si, Gyungjoo Lee, Il Hyun Lee, Ju-Young Lee

**Affiliations:** 1College of Nursing, The Catholic University of Korea, Seoul 06591, Republic of Korea; tldmsdk@catholic.ac.kr (E.S.); kjdooly@catholic.ac.kr (G.L.); 2StatEdu Institute of Statistics, Iksan-si 54630, Republic of Korea

**Keywords:** adolescents, social networking, self-concept, citizenship, gender

## Abstract

This study examined the significance of digital citizenship and gender in the relationship between social media usage time and self-esteem among adolescents. This cross-sectional study was a secondary analysis using national data on 506 Korean adolescents acquired from the 2020 Korea Media Panel Survey. The data were analyzed using SPSS 23.0 and SPSS PROCESS macro. We found that the moderated moderating effects of digital citizenship and gender on the relationship between social media usage time and self-esteem were significant. Both boys and girls had higher self-esteem in groups with higher digital citizenship than in groups with lower digital citizenship. The relationship between social media usage time and self-esteem for boys was positive in the high digital citizenship group and negative in the low digital citizenship group. Conversely, for girls, the relationship between social media usage time and self-esteem was positive in the low digital citizenship group and negative in the high digital citizenship group. It is important to take a differentiated approach that considers the relationship between digital citizenship and gender to promote healthy digital media use and positive self-esteem.

## 1. Introduction

Self-esteem is a subjective judgment about one’s own values [1,2]. It is an important psychological mechanism that explains an individual’s psychological health and adaptation to changes [2]. Positive self-esteem steers emotions and behaviors in a desirable direction [3]. Negative self-esteem is associated with psychological instability, such as depression, deviant behavior, and serious adolescent problems such as suicide [4]. Self-esteem during adolescence is particularly important because it affects an individual’s psychological well-being, interpersonal relationships, social adjustment, and professional success throughout their lives [3,5].

Social comparison in daily life influences self-esteem and reflected appraisals [1,4]. Nowadays, adolescents socialize online: they communicate, share information, and connect through social media [6,7,8]. Social comparison and reflected appraisals on social media can affect adolescents’ self-esteem [4]. Statistics indicate that 95% of U.S. teens have access to a smartphone, and 45% are online ‘almost constantly’ [6]. Childwise found that social networking and messaging friends and family were the predominant reasons for adolescents aged 15–16 to go online, other than for listening to music [9]. Social relationships on online platforms, where teens spend much time, can significantly impact their self-esteem. However, previous studies yielded controversial results on the relationship between social media use and self-esteem [8]. Some studies report that adolescents show low self-esteem when they spend less time with family or friends and spend more time using social media [10] However, in other studies, the use of social media increases social networking opportunities, and is reported to increase self-esteem [11]. These inconsistent findings imply that social media use does not directly influence self-esteem.

Online activities that young people indulge in are associated with risks and harms, including self-harm, sexual and violent content, as well as positive experiences such as self-expression and self-identity [12]. Kim reported that adolescents were using YouTube as a space for emotional exchange, informal learning, identity construction, and experimentation [7]. As digital natives, today’s adolescents must be able to responsibly control the risks of social media and proactively utilize its benefits. Digital citizenship is defined as an ethical and responsible attitude that utilizes information and content through media in a way that is sensible and considerate of others [13]. Fostering a strong digital citizenship can help to build a sense of healthy self-esteem on social media. Despite this significance, there is a lack of research on how digital citizenship affects the relationship between social media use and self-esteem among adolescents.

Many studies report on gender differences in self-esteem, with men showing higher self-esteem than women [14]. The relationship between internet addiction and self-esteem was stronger in men, with boys being more susceptible to the negative effects of internet addiction [15]. According to Twenge and Martin, adolescent girls reported spending more time on social media, texting, while boys spent more time playing games and on electronic devices [16]. They reported that the association between heavy digital media use and lower psychological well-being was generally stronger for girls than for boys. Gender differences in social media use and self-esteem are widely reported in the literature [17]. These associations may also vary according to the degree of digital citizenship by gender.

Against this backdrop, this study has the following specific objectives:To examine the relationship between social media usage time and self-esteem among adolescents using population-representative Korean media panel data;To identify the moderated moderating effect of digital citizenship and gender in this relationship.

## 2. Materials and Methods

### 2.1. Research Design and Data

This cross-sectional study was a secondary data analysis using national data from the Korea Media Panel Survey to examine the moderated moderating effects of digital citizenship and gender on the relationship between social media usage time and self-esteem.

This study used data from the 2020 Korea Media Panel Survey published by the Korea Information Society Development Institute (KISDI). The 2020 Korea Media Panel Survey, an annual panel survey by the KISDI since 2010, was conducted as a door-to-door survey over 10 weeks from June to July 2020. It included 4,537 households nationwide and 10,800 household members aged six years and older sampled through stratified two-stage probability proportional cluster sampling. For this study, we selected a new sample of digital natives who are the most sensitive to the media landscape. Middle and high school students, who are part of the digital 2.0 generation, are the focus of the analysis. The sample was selected in recognition that the age group utilizing digital devices is getting increasingly younger. Furthermore, given that the dependent variable, self-esteem, can only be answered by those aged 13 or older, we considered those aged 13 to 19 as digital natives. We used the 11th round of data from 2020 because it reflects the changes in daily life due to the COVID-19 pandemic; specifically, social media usage increased as offline face-to-face communication decreased. It also included the main variables of this study: social media usage time, self-esteem, and digital citizenship. Excluding those with missing values for key variable questions, data from 506 adolescents aged between 13 and 19 were used.

### 2.2. Variables

#### 2.2.1. Demographic Variables

This study used gender, education level, religion, and region of residence as basic demographic variables. Education level was categorized into elementary school graduate or less, junior high school graduate or less, high school graduate or less, college graduate or less, and graduate school or more based on the final level of education. Residence areas were categorized into municipal and county areas.

#### 2.2.2. Social Media Usage Time

Social media usage time was measured using the 2020 Media Diary data from the Korea Media Panel Survey. A media diary is a 15 min record of media usage by media usage behavior, connection method, and location for three days to identify an individual’s media usage state accurately and specifically. A media diary, a detailed record of media usage by time of day, provides data close to actual usage by reflecting the simultaneous use of other media. In this study, the three-day average value of social media usage time from the 2020 media diary data was used as a variable representing social media usage time. The social media used by the participants were blogs, mini-homepages, Twitter, KakaoStory, and Facebook.

#### 2.2.3. Self-Esteem

For measuring self-esteem, the Korean media panel utilized the self-esteem scale (SES) developed by Rosenberg [1] and adapted by Byungjae [18]. The questions were scored on a 4-point scale ranging from 1 (never) to 4 (always): “I think I am as valuable as other people”, “I think I have a good character”, “I generally feel like a failure”, “I can do things as well as most other people”, “I do not have much to be proud of”, “I have a positive attitude about myself”, “I am generally satisfied with myself”, “I wish I had more respect for myself”, “I sometimes feel like I am worthless”, and “I sometimes think I am not a good person”. Four of the items were reverse-scored, with higher scores indicating higher self-esteem. In this study, the instrument’s reliability, Cronbach’s α, was 0.74.

#### 2.2.4. Digital Citizenship

Digital citizenship was measured using 37 questions from the Korean Media Panel Survey. The questionnaire was developed by KISDI with reference to the Korea Communications Commission’s (2017) Media Literacy Index Development and Regional Disparity Measurement Survey [19]. It was designed to capture users’ acceptance and understanding of media content and industries. It consists of five sub-factors: the ability to distinguish media content, internet information search activity, thoughts on media and media messages, evaluation of media content, and understanding of media industry and regulation. Each question is answered on a 5-point scale (1 = not at all; 5 = very much). The higher the score, the higher the digital citizenship. In this study, the reliability of the instrument was Cronbach’s α = 0.95.

#### 2.2.5. Ethical Considerations

This study used secondary data from the “2020 Korea Media Panel Survey” by KISDI. It was conducted after obtaining an exemption from review from the university’s Institutional Review Board, where this researcher is affiliated (IRB No.: MC23ZASI0010), because the data did not contain personally identifiable information of the research subjects. Files and materials related to the study will be discarded immediately upon expiration of the study period.

#### 2.2.6. Data Analysis

SPSS version 23 was used to analyze the data. Frequency analysis and descriptive statistics were used to examine general characteristics. Mann–Whitney U test was used for analyzing social media usage time according to general characteristics, and the *t*-test was used for studying the differences in digital citizenship and self-esteem. Correlation analysis examined the relationship between social media usage time, digital citizenship, and self-esteem. Multiple regression analysis was used to examine the effect of social media usage time on self-esteem. Normality was calculated based on skewness and kurtosis. The absolute value of kurtosis of social media usage time was more than 7, which did not meet the normality assumption; therefore, regression analysis was conducted using the bootstrap method. Finally, we used the bootstrap method to sample 50,000 times to examine the moderating effect of gender and digital citizenship and tested the moderated moderating effect of process macro 4.0 Model 3. It was tested at the 0.05 level of significance.

## 3. Results

This section is divided into subheadings. It should provide a concise and precise description of the experimental results, their interpretation, as well as the experimental conclusions that can be drawn.

### 3.1. Differences in Key Variables Based on General Characteristics

Differences in key variables based on general characteristics are presented in Table 1. Regarding gender, 264 (56.4%) were male and 242 (43.6%) were female. A total of 285 (56.4%) were middle school and 221 (43.6%) were high school students. Concerning religion, 377 (74.6%) had no religion and 129 (25.4%) had a religion. Further, 476 (93.9%) lived in municipal areas and 31 (6.1%) in county areas. Regarding general characteristics, social media usage time was significantly different by gender (*p* = 0.007), with females (M = 0.27) using social media more than males (M = 0.14). Digital citizenship was significantly different by school level (*p* < 0.001), with high school students (M = 3.06) having higher digital citizenship than middle school students (M = 2.79) (Table 1).

### 3.2. Descriptive Statistics and Correlation Analysis

The descriptive statistics of each factor are shown in Table 2: social media usage time 0.20 h, digital citizenship 2.94, and self-esteem 2.35. The correlation analysis of the variables showed that self-esteem was positively correlated with digital citizenship (r = 0.34).

### 3.3. The Moderated Moderating Effects of Digital Citizenship and Gender

We conducted a multiple regression analysis of the effect of social media usage time on self-esteem while controlling for school level as a dummy variable, which was significant for the general characteristics. Social media usage time did not have a significant effect on self-esteem (*p* = 0.806) (<Table 3, Step 1>). We tested the moderating effect of digital citizenship on the effect of social media usage time on self-esteem (<Table 3, Step 2>). The moderating effect of digital citizenship (B = 0.01, *p* = 0.806) was insignificant. Conversely, the moderated moderating effects of digital citizenship and gender on the relationship between social media usage time and self-esteem were significant (B = −0.22, *p* = 0.023) (<Table 3, Step 3>). When digital citizenship was low, more time spent on social media was associated with lower self-esteem for boys and higher self-esteem for girls; when digital citizenship was high, more time spent on social media was associated with higher self-esteem for boys and lower self-esteem for girls (Figure 1).

## 4. Discussion

This study was conducted to identify the moderated moderating effects of digital citizenship and gender on the relationship between social media usage time and self-esteem among children and adolescents. According to the findings, the moderated moderating effects of digital citizenship and gender on the relationship between social media usage time and self-esteem were significant. For boys, the relationship between social media usage time and self-esteem was positive in the high digital citizenship group and negative in the low digital citizenship group. For girls, the relationship between social media usage time and self-esteem was positive in the low digital citizenship group and negative in the high digital citizenship group.

We found that the high digital citizenship group had higher self-esteem than the low digital citizenship group for both boys and girls. The results support previous studies that have reported digital citizenship as a factor in increasing self-esteem [13]. According to the social comparison theory, all humans have a fundamental need to assess their own opinions, abilities, and situations, and information from comparisons with others often forms the basis of their evaluations of themselves [20,21]. Social media serves as a kind of “social resume” that shows the version of themselves that adolescents are most satisfied with [22]. This comparison of themselves to others affects adolescents’ self-esteem [8]. High digital citizenship is the ability to critically perceive various social comparison situations online, resulting in an increased self-esteem.

In our study, boys with high digital citizenship had higher self-esteem than those with low digital citizenship, and were also found to have higher self-esteem with more screen time. Conversely, boys with low digital citizenship had lower self-esteem than those with high digital citizenship, and were also found to have lower self-esteem with more screen time. Twenge and Martin’s meta-analysis found that men’s social media use can be associated with higher or lower self-esteem depending on the type of content they are exposed to [16]. Exposure to less attractive content than themselves was associated with higher self-esteem, while exposure to more attractive content than themselves with lower self-esteem. In this study, we measured digital citizenship as the ability to distinguish the content on social media. In this study, boys with high digital citizenship had higher self-esteem even when they spent more time on social media, and boys with low digital citizenship had lower self-esteem with more social media time. This suggests that having a critical view of the content does not lower self-esteem in men, regardless of how much time they spend on social media.

We found that social media usage time and self-esteem were negatively related in the high digital citizenship group and positively related in the low digital citizenship group for girls. Unlike boys, higher digital citizenship was associated with higher self-esteem, but higher usage time was associated with lower self-esteem among girls. Conversely, when digital citizenship was low, self-esteem was higher as usage time increased. Previous research reported that adolescents engage in social media to connect with friends and that positive feedback from close friends increases their self-esteem [8,17]. In particular, for girls, social media use is reported to increase self-esteem [23]. Social connections on social media may be more positively and strongly associated with women’s self-esteem than men. Accordingly, the group of girls with low digital citizenship and low self-esteem may increase their self-esteem due to social solidarity in social media. Conversely, girls with high digital citizenship may have high self-esteem but become self-deprecating due to critical thinking about their participation in social media. Nevertheless, it should not be overlooked that girls also had higher self-esteem when digital citizenship was high.

This suggests that the moderated moderating effects of digital citizenship and gender must be considered in the relationship between time spent on social media and self-esteem. For boys, fostering digital citizenship may be a way to increase self-esteem regardless of time spent on social media. Girls also reported higher self-esteem on average among those with high digital citizenship than those with low digital citizenship. Still, the mechanisms by which girls address self-esteem are likely more complex than those for boys. Adolescent boys and girls are reported to have different primary digital media activities. Girls spend more time on social media, and boys spend more time playing games [16]. This study also found that girls spent more time on social media than boys did. Social media usage time may be more relevant to women. The tools used in this study, in particular, determine the right or wrong of content, such as the ability to categorize media content and the level of understanding of the media industry and regulations. Thus, critical thinking about media can lead to a negative interpretation of social media participation, encouraging girls to view social media participation negatively. Therefore, there is a need for further research on factors other than digital citizenship that may affect girls’ self-esteem on social media, including the nature of their social relationships. It is also necessary to organize educational contents that consider gender differences to foster digital citizenship effectively.

This study found that the moderated moderating effects of digital citizenship and gender on the relationship between social media usage time and self-esteem were significant. The data used in this study were collected during the COVID-19 pandemic, which began in early 2020, and is meaningful in that it reflects well the changed daily life in which time spent on SNS increased as offline face-to-face communication decreased. In addition, the study’s focus on identifying individual characteristics that reduce negative social media use among adolescents may help to guide targeted prevention programs. However, there are several limitations that need to be acknowledged. This cross-sectional analysis uses secondary data, limiting the interpretation of causal relationships between variables. Therefore, it is necessary to investigate the causal relationship between social media usage time and self-esteem, digital citizenship, and gender longitudinally. It should also be noted that the construct of digital citizenship is limited due to the limitations of secondary data. Among the limited demographic data provided by the “2020 Korea Media Panel Survey” by KISDI, demographic data of adolescents in this study included gender and education, and socio-cultural background included religion and region of residence. Due to limitations in secondary data, more general characteristics could not be controlled, and this needs to be considered in future research.

## 5. Conclusions

This study’s significance lies in identifying a differential relationship between time spent on social media and self-esteem according to digital citizenship and gender. In designing interventions for promoting healthy digital media use and self-esteem, it is important to take the differentiated approach that considers the effect of digital citizenship and gender on the relationship between digital media use and self-esteem. To ensure healthy digital media use among adolescents, more future research should be conducted to identify specifiers that differentiate boys’ and girls’ self-esteem and digital citizenship. Based on the information presented in this study, we will provide basic information for developing the competencies and education programs necessary to foster digital citizenship in youths, the most active users of digital information. This suggests that the development and education of such educational programs should make integrated efforts to foster balanced digital citizenship not only in schools but also at home and in society.

## Figures and Tables

**Figure 1 children-10-01561-f001:**
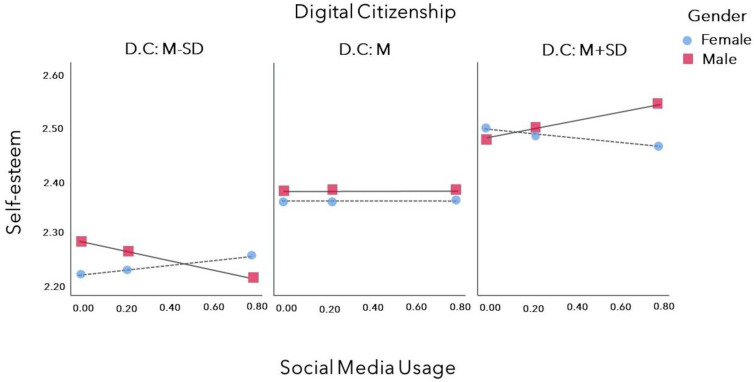
Moderated moderating effects of digital citizenship and gender on the relationship between social media usage and self-esteem. Note: D.C; digital citizenship, M: mean, SD: standard deviation.

**Table 1 children-10-01561-t001:** Differences in key variables based on general characteristics.

Variables	Categories	n (%)	Social Media Usage (hour)			Digital Citizenship			Self-Esteem		
			M	SD	t (*p*)	M	SD	t (*p*)	M	SD	t (*p*)
Gender	Male	264(56.4)	0.14	0.36	−1.75 (0.007 ^†^)	2.88	0.59	−1.13 (0.256)	2.35	0.33	0.17 (0.858)
	Female	242(43.6)	0.27	0.65		2.94	0.54		2.34	0.35	
Education	Middle school	285(56.4)	0.21	0.56	0.18 (0.857)	2.79	0.58	−5.56 (<0.001 ^†^)	2.34	0.36	−0.81 (0.415)
	High school	221(43.6)	0.20	0.47		3.06	0.51		2.36	0.31	
Religion	Yes	129(25.4)	0.18	0.58	−0.59 (0.556)	2.84	0.61	−1.54 (0.123)	2.32	0.33	−1.16 (0.245)
	No	377(74.6)	0.21	0.50		2.93	0.55		2.36	0.34	
Region of living	City	476(93.9)	0.20	0.52	−0.22 (0.822)	2.91	0.56	1.07 (0.281)	2.35	0.34	0.29 (0.765)
	Town	31(6.1)	0.22	0.52		2.80	0.62		2.33	0.35	

^†^: Mann–Whitney test.

**Table 2 children-10-01561-t002:** Correlations.

	M	SD	Skewness	Kurtosis	X	Mo	Y
Social Media usage (X) (hour)	0.20	0.52	5.83	53.30	1.00		
Digital Citizenship (Mo)	2.94	0.57	−0.02	−0.33	0.02 (0.571)	1.00	
Self-esteem (Y)	2.35	0.34	0.44	1.40	0.01 (0.825)	0.34 (<0.001)	1.00

r (*p*).

**Table 3 children-10-01561-t003:** The moderating effects of digital citizenship and gender.

	Step 1			Step 2			Step 3		
	B	SE	*p*	B	SE	*p*	B	SE	*p*
Constant	2.35	0.03	<0.001	1.73	0.07	<0.001	2.08	0.21	<0.001
Gender (male)	0.01	0.03	0.877	0.02	0.02	0.401	−0.22	0.13	0.102
Education (middle school)	−0.03	0.03	0.421	0.04	0.05	0.077	0.04	0.02	0.059
Social Media usage (hour)	0.01	0.03	0.806	−0.03	0.14	0.821	1.09	0.52	0.037
Digital citizenship (Mo)				0.20	0.02	<0.001	0.10	0.07	0.160
Social Media usage × Gender							0.65	0.30	0.031
Social Media usage × Mo				0.01	0.04	0.806	0.37	0.16	0.026
Gender × Mo							0.06	0.04	0.129
Social Media usage × Mo × Gender							−0.22	0.09	0.023
F(*p*) *R^2^*	0.35 (0.699) 0.001			18.19 (<0.001) 0.116			12.16 (<0.001) 0.123		

## Data Availability

The data described in this article are openly available at https://stat.kisdi.re.kr/kor/contents/ContentsList.html?rootId=2010002&subject=MICRO10&sub_div=D&menuId=2010126 (accessed on 6 August 2023).

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
