# Peer review of "The Significance of Digital Citizenship and Gender in the Relationship between Social Media Usage Time and Self-Esteem among Adolescents: A Secondary Analysis"

_children, 2023, doi:10.3390/children10091561_

Round 1
Reviewer 1 Report
The paper discusses the citizenship digital education approach about the social media usage. A major revision is required.
Strengths: the discussed topic is interesting.
Points of weakness: short discussion about self-esteem (par. 2.3.3), short description of the data model and about data significance, data analytics approach, short description of the correlation analysis.
Actions to do:
According to the weaknesses, I suggest to improve the paper by answering to these points:
· The authors should describe deeply the correlation between self-esteem and social media;
· Please provide more details about data model and data model (religion and religion living attribute are interesting).
· Conclusion section should be improved;
· Please summarize better all the results enhancing COVID 19 effects (it is better to propose also a discussion related a period pre-covid 19);
Minor remarks:
In the abstract change the sentence including the words “moderated moderating” (also in the title sounds no well).
English is ok
Reviewer 2 Report
AUTHORS examined the moderated moderating effect of digital citizenship and gender in the relationship between Social Media usage time and self-esteem among adolescents. This cross-sectional study was a secondary analysis using national data on 506 Korean adolescents acquired from the 2020 Korea Media Panel Survey. The data were analyzed using SPSS 23.0 and SPSS PROCESS macro.
AUTHORS found that:- the moderated moderating effects of digital citizenship and gender on the relationship between Social Media usage time and self-esteem were significant.
-Both boys and girls had higher self-esteem in groups with higher digital citizenship than in groups with lower digital citizenship.
-The relationship between Social Media usage time and self-esteem for boys was positive in the high digital citizenship group and negative in the low digital citizenship group.
-Conversely, for girls, the relationship between Social Media usage time and self-esteem was positive in the low digital citizenship group and negative in the high digital citizenship group.
AUTHORS concluded that it is important to take a differentiated approach that considers the relationship between digital citizenship and gender to promote healthy digital media use and positive self-esteem.
This is an interesting study.
I have some minor comments with a pure academic spirit.
1) The purpose must be more effective. Explicit it using the bullet points if needed
2) Avoid short paragraphs (see the 2.1)
3) “3.2. Figures, Tables and Schemes All figures and tables should be cited in the main text as Figure 1, Table 1, etc.” I do not understand, perhaps an error in the texting. However I suggest to describe the figure in details.
Round 2
Reviewer 1 Report
The paper is improved. I suggest to furthermore improve the conclusions
Good
